# A Qualitative and Quantitative Occupational Exposure Risk Assessment to Hazardous Substances during Powder-Bed Fusion Processes in Metal-Additive Manufacturing

Stefano Dugheri [1,*], Giovanni Cappelli [2], Lucia Trevisani [2], Simon Kemble [3], Fabrizio Paone [3], Massimiliano Rigacci [3], Elisabetta Bucaletti [2], Donato Squillaci [2], Nicola Mucci [2] and Giulio Arcangeli [2]

1   Industrial Hygiene and Toxicology Laboratory, Careggi University Hospital, 50134 Florence, Italy
2   Department of Experimental and Clinical Medicine, University of Florence, 50139 Florence, Italy;
    giovanni.cappelli@unifi.it (G.C.); lucia.trevisani@unifi.it (L.T.); elisabetta.bucaletti@unifi.it (E.B.);
    donato.squillaci@unifi.it (D.S.); nicola.mucci@unifi.it (N.M.); giulio.arcangeli@unifi.it (G.A.)
3   Baker Hughes Turbomachinery & Process Solutions, 50127 Florence, Italy;
    simon.kemble@bakerhughes.com (S.K.); fabrizio.paone@bakerhughes.com (F.P.);
    massimiliano.rigacci@bakerhughes.com (M.R.)
*   Correspondence: stefano.dugheri@unifi.it

**Abstract:** Metal-additive manufacturing (AM), particularly the powder-bed fusion (PBF) technique, is undergoing a transition from the short-run production of components to higher-volume manufacturing. The industry's increased production efficiency is paired with a growing awareness of the risks related to the inhalation of very fine metal powders during PBF and AM processes, and there is a pressing need for a ready-to-use approach to assess the risks and the occupational exposure to these very final metal powders. This article presents a study conducted in an AM facility, which was conducted with the aim to propose a solution to monitor incidental airborne particle emissions during metal AM by setting up an analytical network for a tailored approach to risk assessment. Quantitative data about the respirable and inhalable particle and metal content were obtained by gravimetric and ICP-MS analyses. In addition, the concentrations of airborne particles (10–300 nm) were investigated using a direct reading instrument. A qualitative approach for risk assessment was fulfilled using control banding Nanotool v2.0. The results show that the operations in the AM facility are in line with exposure limit levels for both micron-sized and nano-sized particles. The particulate observed in the working area contains metals, such as chromium, cobalt, and nickel; thus, biological monitoring is recommended. To manage the risk level observed for all of the tasks during the AM process, containment and the supervision of an occupational safety expert are recommended to manage the risk. This study represents a useful tool that can be used to carry out a static evaluation of the risk and exposure to potentially harmful very fine metal powders in AM; however, due to the continuous innovations in this field, a dynamic approach could represent an interesting future perspective for occupational safety.

**Keywords:** metal additive; nanoparticle; occupational safety

## 1. Introduction

One of the crucial themes of using green technology to deal with finite resources and global warming is the sustainability of the manufacturing process [1,2]. Direct metal-additive manufacturing (DMAM) produces three-dimensional (3D) metallic objects with geometrical complexity via the layer-by-layer deposition of powder in computerised design and manufacturing schemes [3].

Metal-additive manufacturing (AM) also allows manufacturing to be carried out with advanced materials, such as the cobalt–chromium ceramic alloy [4]. The DMAM process has shown two encouraging merits: good performance and environmental benefits in the

applied parts because of new geometries and the development of the buy-to-fly ratio to manufacture metallic parts. Therefore, DMAM can provide a solution to increase the sustainability of the manufacturing process [5–7], as proposed by the Additive Manufacturer Green Trade Association (AMGTA) (Hollywood, CA, USA), a global no-profit trade group created to promote the green benefits of additive manufacturing over traditional methods. The interest in DMAM processes has steadily increased in industry and in research areas, with the intention to improve the sustainability of the manufacturing process thanks to the possibilities that DMAM offer, such as redesigning products, improving efficiency, and extending the life of a product [6].

To navigate this modern scenario, a detailed understanding of the different AM technologies and their capabilities and impact on safety is essential. Technical committees such as the International Organization for Standardization (ISO) 261 and the American Society for Testing and Materials (ASTM) F42 have classified widely used AM technologies into seven categories [8,9]. Focusing on metal AM, powder bed fusion (PBF) technologies made up 54% of the market in 2020, followed by material/binder jetting (BJ) and direct energy deposition (DED) technologies (16%), material extrusion technologies (10%), and sheet lamination (SL) and VAT photopolymerization technologies (2%) [7]. Other AM processes have been tested for metal fabrication, such as arc welding [10]; however, they are still under investigation and in development, and they are not commercially available yet.

Focusing on PBF, it uses a high-energy power source to melt or sinter metallic powder beds ranging in size from 20 to 60 μm. This technology can be divided into two additional techniques: electron beam melting (EBM), which uses an electron beam, and selective laser melting (SLM), which uses a high-intensity laser. The last one includes selective laser sintering (SLS) and direct metal laser sintering (DMLS) [7,11–13]. The most commonly used feedstock forms of metal for these techniques are powder and wire [14], although other feedstock forms (such as sheets, foils, and slurry) have also been proposed [15–17].

Metal AM is characterised by highly dynamic technological and market development. According to Technavio, incremental growth amounting to USD 4.42 billion is expected in the metal AM market during the period of 2020–2024 [18]. In 2019, the size of the global metal AM market was valued at EUR 2.02 billion, and it is expected to grow at a compound annual growth rate (CAGR) of 27.8% from 2020 to 2027 [19,20]. Nowadays, over 130 metal AM system suppliers are present on the market and are mainly located in the USA (64.4%), China, and Germany [20,21].

The introduction and increasing applications of AM techniques have highlighted the uncertainties about the safety of AM operators. With the current state of technology, only the printing process is automated, while the work steps that take place during the pre- and post-processes are performed manually or semi-automatically. Thus, there are concerns about operators being exposed to metal powder when handling it or when using the AM printers. The metal components of the powder could represent a potential hazard to the health of AM operators [22], particularly in the presence of cobalt. This metal has been assumed to be neurotoxic [23] and may induce cancer [24] as well as lung complications [25,26]. Furthermore, during the metal AM process, nano-sized particles can be generated, and nano-scaled particles have been suggested to possess different toxicological properties than larger particles and may have high cellular toxicity [27]. Ultrafine particles (UFPs), which are naturally occurring nanomaterials, are ambient particulate matter (PM0.1) and contain nano-scale particles with a diameter of <0.1 μm [28].

In the everyday environment, we encounter natural (e.g., sea salt, ash, etc.), accidental (e.g., products resulting from technological processes, combustion exhaust), and engineered nanoparticles and UFPs. The average background concentration of nanoparticles in a clean indoor environment ranges from 1000 to 10,000 particles/cm$^3$ [29], while out-door urban air contains between 10,000 and 50,000 particles/cm$^3$ [30].

Concerning metal nanoparticles, the health effects that they can induce are well-known hazards in other metal processing activities, such as in welding [31,32]. Nanomaterials with a diameter of <0.1 μm can become deposited in the upper airways and in the pulmonary

alveoli [33,34]. Moreover, due to their size, these nano-objects can spread into the bloodstream, cross biological membranes, amass in organs, and pass through the blood–brain barrier [35–41]. Living cells can internalize these nanoparticles, resulting in processes such as metabolism, proliferation, differentiation, or lysis being dysregulated [42]. In addition, macrophages do not recognise nanoparticles, resulting in a potential inflammatory microenvironment [43].

Nowadays, no specific international regulations nor occupational exposure limits have been proposed for nanoparticles by the European Community. The World Health Organization (WHO) has proposed guidelines and recommendations to manage exposure to nanoparticles. In 2018, 56 occupational exposure limits were reported by the American Conference of Governmental Industrial Hygienists [44]. Among these, the US NIOSH sets a 0.3 mg/m$^3$ 10 h time-weighted average (TWA) as the recommended exposure limit (REL) for TiO$_2$ nano-objects and an REL of 1.0 μg/m$^3$ for carbon nanotubes [45]. In Germany, the value of 0.3 mg/m$^3$ has also been proposed by the Deutsche Forschungsgemeinschaft (DFG) as a reduced general limit value for respirable dust. The Nanosafety Research Centre of the Finnish Institute of Occupational Health (FIOH) has defined a target value of 20,000 nanoparticles/cm$^3$ (at a density > 6000 kg/m$^3$) for an exposure time of 8 h [46]. This value was also later assumed by the Institute for Occupational Safety and Health of the German Social Accident Insurance (IFA DGUV) and the IVAM Environmental Research UVA BV in the Netherlands [47,48].

There are ongoing substantial efforts to develop risk assessment methods, particularly for nanoparticles, due to the lack of occupational exposure limits (OELs) and standardised sampling strategies. Grieger et al. [49] suggested simplified risk assessment schemes that could be quickly developed and applied to scenarios with a risk of nanoparticle inhalation. Simplified risk evaluation schemes could represent a ready-to-use alter-native to chemical risk assessment and could act as a pre-evaluation to hierarchically organize the risks related to nanomaterials and the tasks that they are involved in.

Legislative decree 81/2008 and its subsequent integrations and modifications is the current norm in Italy; it reorganizes and updates all of the provisions related to health and safety at work. It also provides a subsequent definition: ""risk" means the probability of achieving the potential level of damage under conditions of use or exposure to a particular factor or agent or to their combination". Risk is often expressed in terms of the consequences of an event and the associated likelihood of occurrence. The terms of consequences represent the severity, while the likelihood of occurrence is the probability. As indicated in D. Lgs. 81/08, risk assessment is mandatory and is thus a crucial step in the health and safety mindset.

Many different screening risk assessment methods for chemical hazards exist [50], and they principally focus on occupational risks to human health and less on environmental ones. The three basic variants of these screening tools, which are the basis for the development of more complex methods, are risk ranking (RK) [51], chemical ranking and scoring (CRS) [52], and control banding (CB). Generally, these three basic tools are employed in different domains due to their different approaches to the hazard parameters and their related scoring and ranking. CB is specifically applied to manage occupational chemical risks: it is a tool to manage the risks resulting from exposure to potentially hazardous substances in the absence of firm toxicological and exposure data. It is based on the risk paradigm, where risk is a function of the severity of impact (hazard) and the anticipated probability of that impact (exposure) [53]. Risk evaluation methods are more straightforward than other methods concerning nanoparticles and are only based on a binary yes/no scale and do not include complicated aggregation algorithms [54]. Among these, the CB Nanotool v2.0 method developed by Zalk et al. [55] is the most frequently used method.

In an environment where technological evolution and operating procedures are constantly changing, a static risk assessment may not be sufficient or may rapidly become inadequate [56]. As a result, various studies have proposed the evolution of the classic two-dimensional approach of the risk matrix to obtain a dynamic assessment that can represent

the best solution. This is achieved through machine learning applications: a computer can be trained to assess risk by processing a large amount of information in the form of indicators from normal operations and past unwanted events; once the model has learned how to categories risk, it uses its knowledge to assess risk from the state of the monitored system in real time [57]. Moreover, other studies have focused on the development of new dynamic approaches for risk management to propose solutions that can also fit emergencies and unexpected accidents [58,59].

The rapid increase in the production, consumption, and pollution of nanoparticles has led to the need for methods that can be used to assess the risks related to nanomaterials. To our knowledge, only a few publications [60–63] have dealt with qualitative and quantitative risk assessment in occupational exposure to hazardous substances during metal AD processes by considering the whole scenario, even the dermal and biological monitoring. This study aims to characterize metal particle emissions during metal AM operations and to obtain information about correct measuring methods to identify appropriate preventive procedures using different measuring techniques that have been optimised for different sizes of airborne particulate matter. This publication presents measured data on PBF welding with nickel-, chromium-, and cobalt-based alloys observed inside the new Additive Laboratory of the Baker Hughes plant in Florence, Italy. In recent years, the Florence plant focused on Turbomachinery and Process Solutions and was profuse in developing AM applications in the oil and gas industry. Indeed, this industrial centre represents the core of excellence for the research, development, and manufacturing of gas turbines, compressors, pumps, valves, and services. During the study, we observed the difficulties that users face in obtaining independent and reliable data on the safe use AM for metal applications. Our goal is to assess exposure using biological monitoring, environmental monitoring, and previous qualitative evaluation methods to assist in improving safety and air quality in the metal AM workplace. Our new, integrated monitoring approach is described here to provide a protocol to evaluate the chemical risks involved in AM oil and gas industrial workplace activities. Moreover, the evaluation tools applied to the scenario of the Additive Laboratory in this study could be used as an exportable risk assessment database for metal AM activities in other structures or fields.

## 2. Materials and Methods

The monitoring campaign was carried out at the new Additive Laboratory of the Baker Hughes plant in Florence over the work shifts of three consecutive days. Initially, a site inspection was carried out to evaluate the conditioning system and all of the operative procedures; then, the occupational exposure was evaluated as follows:

- A gravimetric quantitative assessment of the exposure to particulate matter;
- A quantitative assessment of the exposure to UFPs during a particular AM task by direct reading measurements;
- A quantitative evaluation determining the dermal exposure of the workers during AM operations;
- A qualitative risk assessment using CB.

### 2.1. Facility and Process

The monitored manufacturing area was characterised by the presence of seven powder-bed fusion machines from different manufacturers for the AM of high-tech metallic components with different powder compositions. In this study, alloys with particle sizes within the range from 15 to 65 μm and containing high levels of cobalt, chromium, and nickel were used. The manufacturing area was approximately 400 m$^2$ and 6.5 m in height. Eleven air-supply diffusers are located on the ceiling of the manufacturing room, while seven air extractors are positioned at floor level (Figure 1).

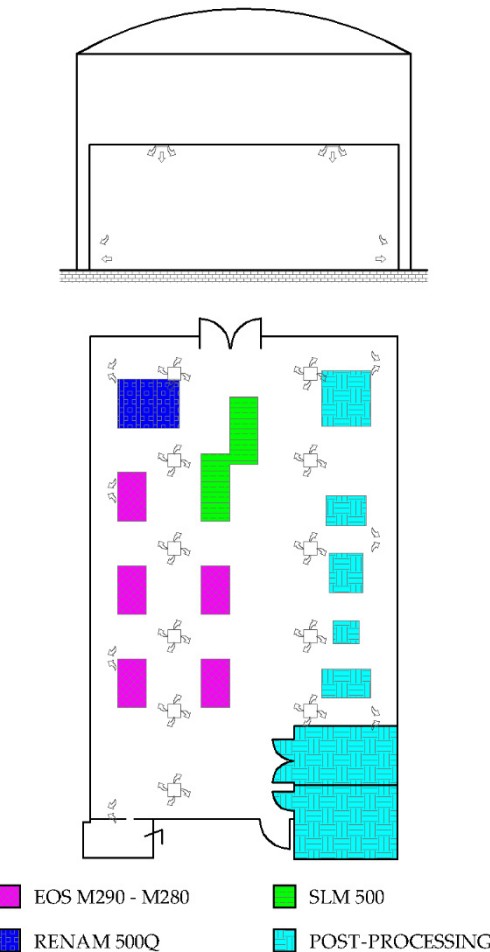

**Figure 1.** Graphic representation of the metal-additive area.

The volume of air supplied to the manufacturing area is equal to 13,000 m$^3$/h, with an air velocity of 2.4 m/s, while the volume of the extracted air is equal to 10,000 m$^3$/h at 2.6 m/h, according to the manufacturer. Based on the room volume and exhaust airflow rate of the manufacturing area, the air changes per hour were calculated, resulting in an average of 7.6 air changes per hour. The general ventilation system is equipped with corrugated filter cells in synthetic fibre (class ISO Coarse 55%—ISO 16890), which act as a pre-filter module, and class H13 HEPA filters (EN 1822:2010), which act as filter units for fine particulate. The filtering units are cleaned and checked monthly. Differential pressure switches are installed on filtering sections, and they are linked to a centralised control system that acquires data continuously: when a difference between the measured pressure and project setting pressure is revealed by the central system, an alarm is activated, and the filter unit is replaced. The manufacturing area is maintained below 24 °C, and the relative humidity is ≤65%. The printing machines are in two parallel "lines" in the same production hall, with one line comprising 1 EOS M290 (EOS GmbH, Krailling, Germany), 1 EOS M280, and 1 SLM-500 (SLM-Solutions, Lubeck, Germany) and the other comprising 1 EOS M280, 2 EOS M290, and 1 renAM 500 Q (Renishaw, Wotton-under-Edge, UK) (Figure 1). The PBF process used by these printers was described well by Graff et al. [61]: the building plate is covered with a uniform thin layer of metal powder; then, this layer is fully melted by a laser before the addition of a new layer of metal powder; this process is repeated until the entire structure is completed. This process is performed in a confined chamber filled with argon as an inert gas. At the end of the printing process, the unused powder is recovered with an external specific vacuum cleaner or with an on-board system that is part of the printers (a perforated grid that drains the powder manually). Then, the powder that has been recovered with an external vacuum system needs to be sifted to be reused in production.

These AM machines are program-controlled, industrial standard systems; still, all of these actions require the powder to be handled, such as while filling the internal storage tank of the printer with the powder or while vacuuming the finished parts, and these tasks are performed manually. All of the post-process steps, such as the deburring, the full-depowdering, and the grinding of the printed structure, are carried out manually in a restricted and separate area with an additional air-supplier system and operational protocols. In the safety regulations for the manufacturing area, there are instructions for the operators regarding protective clothing that has been designed for nanoparticle exposure. Particularly, every time powder is being handled, a red light is turned on, and all of the operators must wear personal protective equipment (PPE), such as powered air-purifying respirators with a P3 filter and gloves, and, during critical operations (such as while refilling the powder manually or depowdering the manufacture with compressed air), a Tyvek suit (DuPont, Wilmington, DE, USA).

On three consecutive days in September 2021, repeated work cycles were evaluated using personal air sampling and direct monitoring devices. The operations that were monitored were:

- The refilling the machine tank with powder both using the manual open system (O1) and the closed one (O2);
- The removal and cleaning the final product and the recovery of unused powder (inside the machine operating area) with both an external specific vacuum cleaner (O3) and by manually dragging the powder in the perforated grill around the building plate (O4);
- The sifting the recovered powder (O5);
- The depowdering of the final product with compressed air in the post-process restricted area (O6).

### 2.1.1. Exposure Quantitative Assessment

A conventional sampling of the particles was performed on four different operators using a pre-weighted 25 mm, 0.8 μm pore mixed cellulose ester filter to compare the results with the occupational exposure limits for the inhalable and respirable fractions. The inhalable and respirable dust concentrations were measured by means of personal sampling over full 4 or 8 h shifts using IOM and Higgins cyclones at a flow rate of 2.0 L/min and 2.2 L/min, respectively. The sampling devices were installed within the individual breathing zone of each worker. Filters were previously conditioned in temperature- and humidity-controlled environment for 48 h and were then weighed; the same procedure was carried out after the sampling. Gravimetric determination was performed to quantify concentrations of the airborne inhalable and respirable dust using a Climatic Cabinet Sartorius SCC400L (Sartorius Italy S.r.l., Varedo, Italy). The metal identities were analysed with an iCAP Q ICP-MS (Thermo Fisher Scientific, Waltham, MA, USA).

For the nanoparticle measurements, a MiniWras (MW) (GRIMM Aerosol Technik Airring GmbH & Co., Airning, Germany) was employed. This device is a portable instrument that permits the simultaneous and real-time monitoring of micron- and nanoparticles by combining optical and electrical particle detection. It also has other features, such as being able to determine the lung-deposited surface area (LDSA) or the "active" surface area ($\mu m^2/cm^3$) concentration, providing simultaneous PM1, PM2.5, and PM10 measurements, and having a 10 nm to 35 μm ultra-wide particle size range, 41 high-resolution particle size channels, remote data transmission, and instrument control. The MW was positioned in the working area of the operator around the printing machine during the monitored tasks.

Potential dermal exposure to metal nanoparticles was evaluated using an 0.8 μm pore mixed cellulose ester (MCE) filter as a pad test, and the filters were attached to the operators' arms, chest, and front side of the shoulders; moreover, biological monitoring results, such as the urinary concentrations of metals in 2021 and in previous years, were also evaluated.

To analyse metals in dust samples, a new method was developed based on previous experience and thanks to collaboration with instrumentation specialists. Three clean MCE filters were spiked at a low, intermediate, and high concentrations to verify that

the analytes had been analysed correctly. The MCE filters were digested with nitric acid using ultraWAVE microwave digestion systems from Milestone Srl (Sorisole, Italy). Then, the digested solution was diluted to achieve an acid concentration of less than 5% before in the solution was analysed in ICP-MS. Multielement calibration standards and continuing calibration verification solutions were prepared to have the same acid concentration as the samples. The ICP-MS analysis was carried out in collision mode using helium as a cell gas and using kinetic energy discrimination (KED) to perform a multielement analysis at low levels.

The Baker Hughes personnel who were exposed to metal powder in the additive laboratories of the Florence plant had been under monitoring since 2010; urine analysis was always carried out by an external laboratory.

### 2.1.2. Risk Qualitative Assessment

Control Banding Nanotool v2.0 (CBN) was applied to qualitatively assess the risk associated with nanoparticles in the monitored operations. In other studies, CBN has shown the potential to assess occupational exposure to incidental nanoparticles [63] even if it was originally created to assess the risks associated with engineered nanomaterials. CBN assigns severity and probability scores to every task, determining the risk level using a four-by-four matrix (Table 1) [64].

**Table 1.** Risk level matrix.

|  |  | Probability | | | |
|---|---|---|---|---|---|
|  |  | Extremely Unlikely (0–25) | Less Likely (26–50) | Likely (51–75) | Probable (76–100) |
| Severity | Very high (76–100) | RL3 | RL3 | RL4 | RL4 |
|  | High (51–75) | RL2 | RL2 | RL3 | RL4 |
|  | Medium (26–50) | RL1 | RL1 | RL2 | RL3 |
|  | Low (0–25) | RL1 | RL1 | RL1 | RL2 |

There are four risk levels (RLs) that are identified by the CBN; each one is linked to a control band, with RL1 corresponding to general ventilation, RL2 corresponding to fume hoods or local exhaust ventilation, RL3 corresponding to containment, and RL4 corresponding to the need to seek specialist advice.

The features of the nanomaterials determine the severity score, and both parental materials (PM) (such as fresh powder or powder generated by the producers) and incidental nanoparticles (IN) are created during the printing processes. The PM and IN factors are listed in Tables 2 and 3.

The overall severity scores can be categorised as follows: low severity (0–25); medium severity (26–50); high severity (51–75); and very high severity (76–100).

The probability score is based on aspects related to exposure (Table 4). These aspects determine the extent to which employees may be potentially exposed to nanoscale materials. Essentially, the probability score is based on the potential for nanoparticles to become airborne.

The overall probability score is based on the sum of all the points from the probability factors and ranges from 0 to 100. It can be categorised into four tiers: a score from 0 to 25 is extremely unlikely; a score from 26 to 50 is less likely; a score from 51 to 75 is likely; and a score from 76 to 100 is probable.

**Table 2.** Severity scores for parental materials.

| | Occupational Exposure Limits (OELs) | | | | |
| --- | --- | --- | --- | --- | --- |
| | **<10 μg/m³** | **10–100 μg/m³** | **101–1000 μg/m³** | **>1000 μg/m³** | **Unknown** |
| Points | 10 | 5 | 2.5 | 0 | 7.5 |
| | | Carcinogenicity | | | |
| | Yes | No | | Unknown | |
| Points | 4 | 0 | | 3 | |
| | | Reproductive Toxicity | | | |
| | Yes | No | | Unknown | |
| Points | 4 | 0 | | 3 | |
| | | Mutagenicity | | | |
| | Yes | No | | Unknown | |
| Points | 4 | 0 | | 3 | |
| | | Dermal Toxicity | | | |
| | Yes | No | | Unknown | |
| Points | 4 | 0 | | 3 | |
| | | Asthmagen | | | |
| | Yes | No | | Unknown | |
| Points | 4 | 0 | | 3 | |

**Table 3.** Severity scores for incidental nanoparticles.

| | Surface Chemistry | | | |
| --- | --- | --- | --- | --- |
| | **High** | **Medium** | **Low** | **Unknown** |
| Points | 10 | 5 | 0 | 7.5 |
| | | Particle Shape | | |
| | Tubular/Fibrous | Anisotropic | Compact/Spherical | Unknown |
| Points | 10 | 5 | 0 | 7.5 |
| | | Particle Diameter | | |
| | 1–10 nm | 11–40 nm | >40 nm | Unknown |
| Points | 10 | 5 | 0 | 7.5 |
| | | Solubility | | |
| | Insoluble | Soluble | Unknown | |
| Points | 10 | 5 | 7.5 | |
| | | Carcinogenicity | | |
| | Yes | No | Unknown | |
| Points | 6 | 0 | 4.5 | |
| | | Reproductive Toxicity | | |
| | Yes | No | Unknown | |
| Points | 6 | 0 | 4.5 | |
| | | Mutagenicity | | |
| | Yes | No | Unknown | |
| Points | 6 | 0 | 4.5 | |
| | | Dermal Toxicity | | |
| | Yes | No | Unknown | |
| Points | 6 | 0 | 4.5 | |
| | | Asthmagen | | |
| | Yes | No | Unknown | |
| Points | 6 | 0 | 4.5 | |

**Table 4.** Probability scores.

| | Estimated Amount of Material Used | | | |
| | >100 mg | 11–100 mg | 0–10 mg | Unknown |
|---|---|---|---|---|
| Points | 25 | 12.5 | 6.25 | 18.75 |
| | | Dustiness/Mistiness | | |
| | High | Medium | Low | Unknown |
| Points | 30 | 15 | 7.5 | 22.5 |
| | | Numbers of Employees with Similar Exposure | | |
| | >15 | 11–15 | 6–10 | 1–5 | Unknown |
| Points | 15 | 10 | 5 | 0 | 11.25 |
| | | Frequency of Operation | | |
| | Daily | Weekly | Monthly | >Monthly | Unknown |
| | 15 | 10 | 5 | 0 | 11.25 |
| | | Duration of Operation | | |
| | >4 h | 1–4 h | 30–60 min | <30 min | Unknown |
| | 15 | 10 | 5 | 0 | 11.25 |

## 3. Results and Discussion

The techniques and instruments involved in metal-additive manufacturing are undergoing a significant and rapid evolution of its techniques and instruments [7]. The wide variety of operations, machines, settings, and different usable metal alloys represent a fascinating challenge for assessing the risks related to AM operations. According to Kaierle et al. [65], risk evaluation in the workplace requires a multi-step strategy that includes obtaining information about processes, such as the steps involved in the process and the materials used in that process, and the individual activities that are carried out by employees. Particularly, based on the behaviour of the hazardous substances, their quantity, and their toxicity, this study on metal AM and potential exposure to nano-sized, or UFP, and micron-sized, or fine particle (FP), materials was carried out and considered relevant working steps for AM and several measuring points using representative sampling (personal and environmental) and standardised measurement and analysis methods. Particularly, the applied instrumentation allows us to appreciate the trends in both micron- and nano-sized particles using the various metrics indicated in UNI EN 16966:2019, such as the particle number ($\#/cm^3$), active surface area ($\mu m^2/cm^3$), and mass ($\mu g/m^3$) or volume ($\mu m^3/m^3$).

### 3.1. Exposure Quantitative Assessment

The filter-based particle measurements that were used to assess the personal occupational exposure to inhalable and respirable dust were performed gravimetrically. The results show that the particle concentrations were within the OELs for all of the monitored operators when applied in all of the evaluated tasks. For the inhalable dust, the concentrations ranged between 0.016 and 1.390 $mg/m^3$, while for respirable dust, the values range from 0.015 to 0.96 $mg/m^3$, as shown in Table 5. The ICP-MS analysis of the filters showed increased levels of chromium, cobalt, and nickel in the AM laboratory, which was determined based on the composition of the used alloy declared by the manufacturer. The pad samples were analysed by ICP-MS, and they showed the presence of chromium, cobalt, and nickel, and they were especially prevalent in the arms pads.

**Table 5.** Inhalable and respirable dust sampled with IOM and Higgins cyclones results.

| N. Samples | Time (min) | Mean. Volume IOM (L) | Inhalable Dust Concentration ($mg/m^3$) | | | Mean. Volume Higgins (L) | Respirable Dust Concentration ($mg/m^3$) | | |
|---|---|---|---|---|---|---|---|---|---|
| | | | Min. | Max. | Mean (s.d.) | | Min. | Max. | Mean (s.d.) |
| 12 | 349 | 698 | 0.016 | 1.390 | 0.363 (0.453) | 767.75 | 0.015 | 0.963 | 0.145 (0.273) |

Concerning the metal nanoparticle exposure assessment, the total particle number concentrations (TPNC) were measured using MiniWras before the beginning of the work-shift to determine the background level (BCK) and during each hazardous task considered; the temporal progression of the TPNC can be used to evaluate the risk related to individual activities that are carried out by the printer operator. The results are shown in Table 6, and the trends of the TPNC for the monitored operations over time are shown in Figures 2–7. In addition, the more relevant size classes of the particles on the TPNC were reported (Table 6) for each task. The trends of LDSA, PM, and inhalable dust as well as the thoracic and respirable concentrations of the monitored tasks are reported in the Supplementary Materials.

**Table 6.** Nanoparticle emissions during the monitored operations in AM and the relative ratio with the background value.

| Operations | OEL (N°/cm³) | Reference Value Indoor (N°/cm³) | Background (N°/cm³) | TPNC (N°/cm³) | Higher Particle Class (nm) | TPNC/BCK |
|---|---|---|---|---|---|---|
| 1 | | | | 2798 | 52–139 | 1.7 |
| 2 | | | | 2111 | 52–139 | 1.3 |
| 3 | 20,000 * | 1000–10,000 ** | 1604 | 3856 | 52–139 | 2.4 |
| 4 | | | | 2689 | 52–139 | 1.7 |
| 5 | | | | 4611 | 52–139 | 2.9 |
| 6 | | | | 10,572 | 52–139 | 6.6 |

\* FIOH-IFA DGUV-IVAM [48]. \*\* Seipenbusch et al. [29].

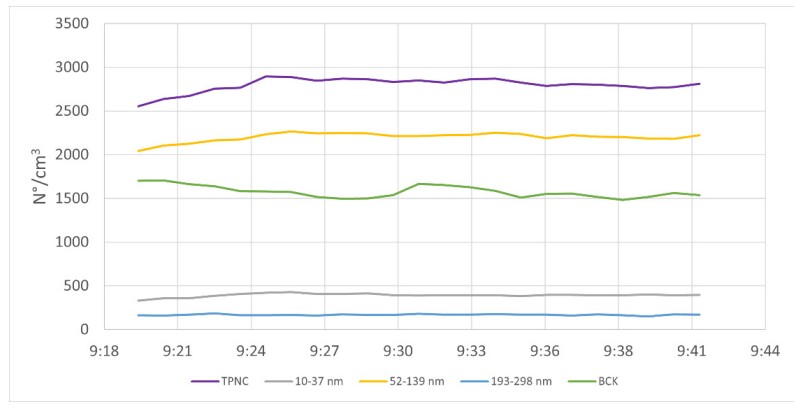

**Figure 2.** TPNC trends and the main particle size classes for O1–manual loading of the metal powder on the EOS 290 M.

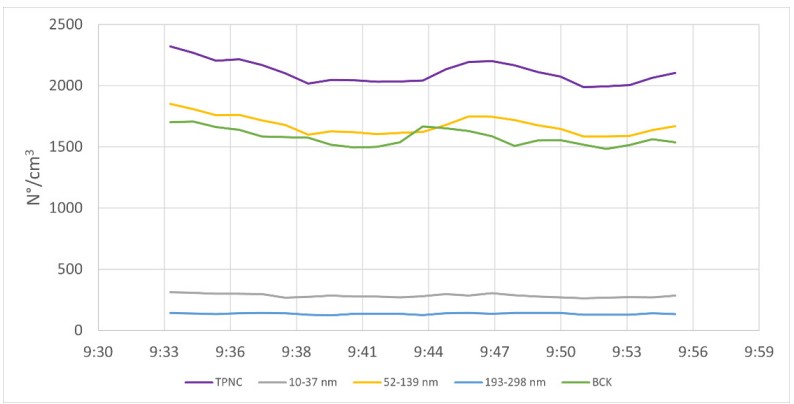

**Figure 3.** TPNC trends and the main particle size classes for O2–loading of the metal powder on the EOS 290 M with a closed system.

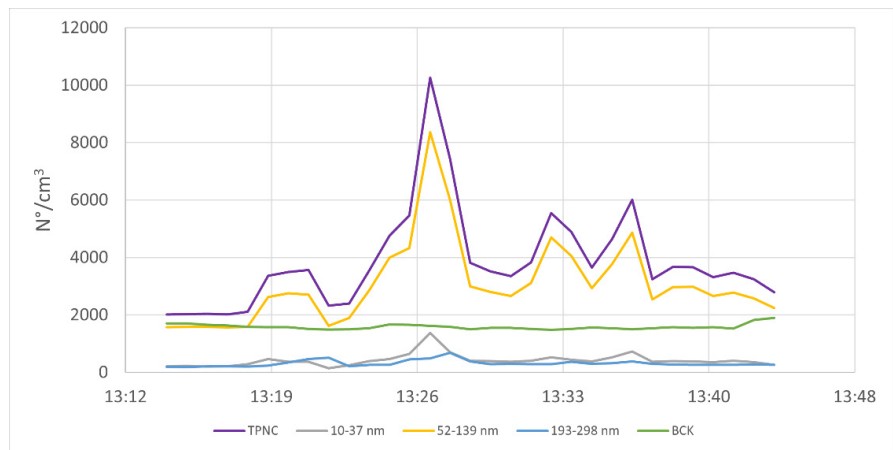

**Figure 4.** TPNC trends and the main particle size classes for O3–removing and cleaning of the final product and the recovering of unused powder with an external specific vacuum cleaner.

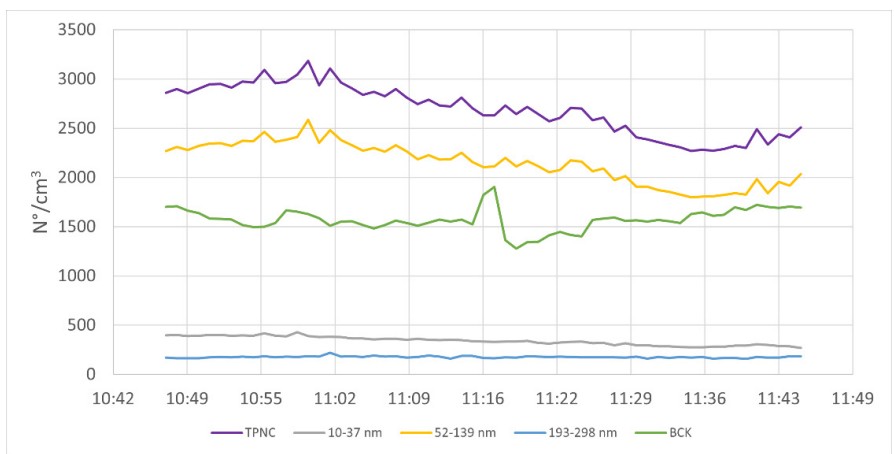

**Figure 5.** TPNC trends and the main particle size classes for O4–removing and cleaning of the final product and the recovery of unused powder by manually dragging the powder into the perforated grill around the building plate.

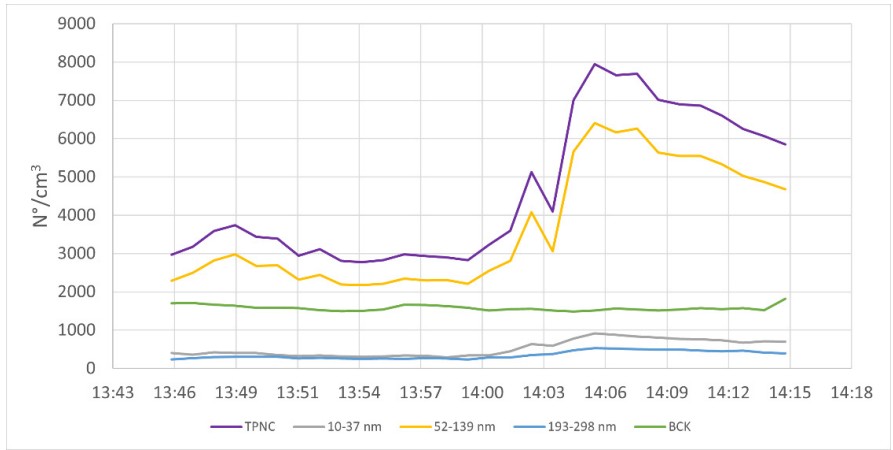

**Figure 6.** TPNC trends and the main particle size classes for O5–sifting of recovered powder.

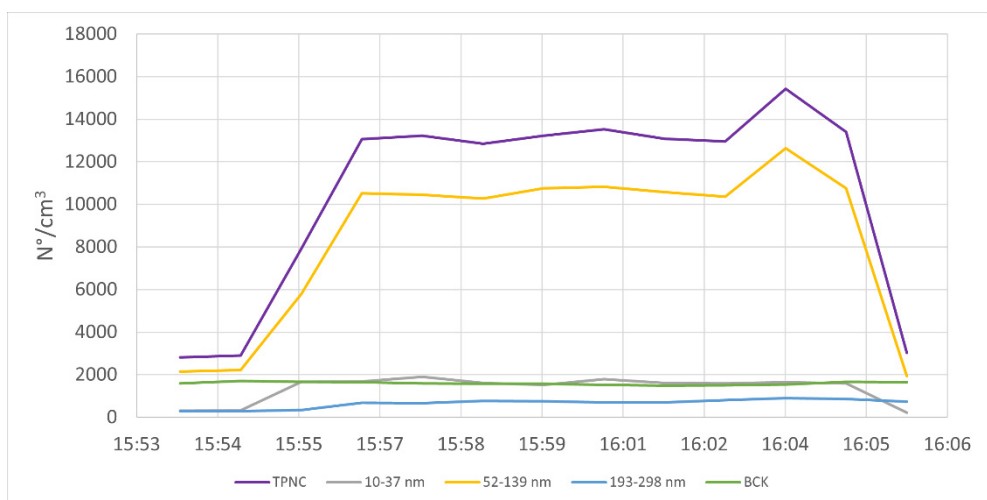

**Figure 7.** TPNC trends and the main particle size classes for O6–depowdering of the final product with compressed air in the post-process restricted area.

The findings of this study concerning the FPs, particularly the inhalable and respirable fractions, showed that over the three days of monitoring, no operators exceeded the respective OEL, in accordance with the results of other similar studies [46,61]. Indeed, Walter et al. [66] and Beisser et al. [60] observed that the AM of high-tech metallic components induced non-relevant concentrations of inhalable and respirable dust (Table 5). Comparatively, Lewinski et al. [60] observed little to no increase in the concentrations of particles in the size range of 0.3–10 μm during standard metal printer operations using a 30 μm metal powder (60% stainless steel and 40% bronze). Regarding the FPs, some authors have carried out studies that have focused on the morphological features, size distribution, and chemical properties of its components [67]. This approach could generate a great deal of crucial information regarding the risks related to the FPs; however, it is not easily applicable, and it is still under study. Furthermore, validated approaches to assess the occupational exposure to FPs, particularly the inhalable and respirable fractions, must face the related OEL; thus, the use of a conventional active personal sampler is required [61].

Regarding UFPs, the increasing interest in occupational safety to nanoparticle exposure has led to the development of cascade impactors. These samplers are characterised by the inclusion of cut-off aerodynamic diameter that covers the nano-size range and include the recently developed micro-orifice uniform deposit impactor (MOUDI), electric low-pressure impactor (ELPI), and NCTU micro-orifice cascade impactor (NMCI) [68]. However, in occupational contexts, it is common to use direct-reading instruments, such as condensation particle counters (CPCs), optical particle counters (OPCs), electrical low-pressure impactors (ELPIs), fast mobility particle sizers (FMPSs), and/or scanning mobility particle sizers (SMPSs). Other strategies use filter-based samples and analyse the collected material by means of scanning electron microscopy (SEM), transmission electron microscopy (TEM)s, energy-dispersive X-ray analysers (EDSs), inductively coupled plasma mass spectrometry (ICP-MS), and X-ray fluorescence (XRF). Different approaches have been proposed and used to study, monitor, and control exposure to metal nanoparticles. TSI (Shoreview, MN, USA) direct reading instruments were proposed to measure nanoparticle concentrations in metal AM processes. Sousa et al. [63] used a CPC Model 3007 with a particle size range of 10 nm to >1 μm and a 1 s time resolution [46], while another group used the SMPS NanoScan Model 3910, which directly measured the number of UFPs as well as the sizes of those ranging from 10 to 420 nm in diameter in 13 size channels via electrical mobility with a sample flow rate of 0.75 L/min and a particle concentration measurement limit from 100 to 1,000,000 particles/cm$^3$ [60]. Dunn et al. [69] used an FMPS Model 3091, which measures particles ranging in from 5.6 to 560 nm through 32 size channels with a fast response rate of 1 s and a high sampling flow rate of 10 L/min while minimizing

the UFP diffusion losses to the tubing surface. The FMPS electrical mobility measurements are similar to those determined by NanoScan; still, the FMPS spectrometer uses multiple low-noise electrometers for particle detection instead of a condensation particle counter [69]. Besides the TSI, other instruments have been applied for UFP evaluation, such as the Nonotracer [61], which detects and counts UFPs from 10 to 300 nm and carries out real-time particle concentration ($N°/cm^3$) and average particle diameter measurements. The geometric surface area of the particles must also be considered because nanoparticles are characterised by a highly developed surface area. Any real-time measuring instrument can determine this nanoparticle parameter; nevertheless, some devices can measure the lung-deposited surface area (LDSA) or "active" surface area, an important metric for determining the adverse health effects of aerosol particles [70–72]. The DiSCmini [73] can measure particles with diameters between 10 and 300 nm and ranging in concentration from 103 to above 106 particles/$cm^3$ with ±30% accuracy, and it can determine the total particle number concentration as well as the LDSA ($\mu m^2/cm^3$) concentration and mean particle diameters. In our study, the MiniWras, which allows for the simultaneous and precise real-time monitoring of both micron- and nanoparticles and is the only portable instrument on the market with this capability, was used for the UFP assessment. It is a state-of-the-art system that fits the purposes of the study, combining an optical aerosol spectrometer and electrical particle detector into one single device with a wide particle size range (from 10 nm to 35 $\mu m$), 41 high-resolution particle size channels, remote data transmission, and instrument control. The MiniWras was used to identify emission sources according to ISO approach ISO/TS 12901-2:2014: the values recorded during the monitored tasks were referred to as the background-level concentration [74]; the lowest exposure level has a ratio below 1.1, so the TPNC changes less than 10% from the background level and is not significant, while the highest exposure level can have a ratio higher than 2, so the TPNC is double the background level (Table 6).

In this study, the measured concentration of UFPs, reported as TPNC, revealed that the metal AM lab did not show a significant increase compared to the background values or welding environments with the exception of the depowdering of the final product with compressed air and the sifting of the used powder, which showed TPNC values that are sevenfold and threefold higher than the background values, respectively. However, the background values are very low and are in alignment with a clean indoor environment (from 1000 to 10,000 particles/$cm^3$) [29]. Moreover, these operations are carried out in a restricted area with supplementary air-supply diffusers and extractors; in addition, the sifting process, after the initial steps, is fully automated, and the operators are able to leave the restricted area until the end of the operation, and concerning the depowdering step, the operators are fully equipped with the PPE mentioned above. The other monitored tasks that are presented here only show slight and insignificant elevations in the TPNC compared to the background value. These surveys are in line with other studies focused on 3D printing. Beisser et al. [60], who evaluated occupational exposure during AM with an aluminium and steel powder, found that the UFPs were similar between ambient air and during the manufacturing process. Other studies [61,63] reported transient peaks of 10–300 nm particles in AM facilities; even our study revealed the main presence of particles with an average size between 52 and 139 nm (Table 6). In contrast to this, other studies showed that the levels and the generation of nanosized particles smaller than 300 nm are very limited during AM operations [46]. This variability could be due to the powder used in the monitored tasks because the new powder being provided by the manufacturer has a defined size; still, after several recoveries from the printing processes and sifting steps, the average particle size could stand to be lower.

This study applied the limit value of 20,000 p/$cm^3$ to assess the occupational exposure to UFPs; however, recently, nanoparticle properties such as the surface area and concentration have been evaluated, meaning that improved exposure metrics have been obtained rather than mass concentration metrics only [75–77]. According to this, safe levels, also known as nano reference values (NRVs), were proposed by the Dutch Social and

Economic Council (SER, Sociaal-Economische Raad): 0.01 fibre/cm$^3$ for carbon nanotube and metal oxide fibres, 20,000 p/cm$^3$ for particles with a density higher than 6 g/cm$^3$ (metallic particles and metal oxides), and 40,000 p/cm$^3$ for particles with a density lower than 6 g/cm$^3$. These values are temporary for particles ranging from 1 to 100 nm and may change according to new information about nanoparticle toxicity.

These more defined limit values are specifically related to engineered nanomaterials; still, occupational risk can often be due to the creation of incidental nanoparticles as the products of various tasks during AM manufacturing. In this scenario, qualitative approaches for assessing the risk of exposure to these particles seem to be an alternative or complementary addition to quantitative analysis [78,79].

### 3.2. Risk Qualitative Assessment

Control banding has been recognised as a suitable algorithm for evaluating and determining the risks of nanoparticles to human health [80], especially when considering that the chemistry, physicochemical properties, concentration, and time-mode of exposure heavily reflect the hazard potential of nanoparticles. Several CB solutions have been proposed [80] to evaluate the occupational risk of nanomaterials using alternative methodologies to classify the prevailing risk using a ranking matrix. Among these, NanoSafer [81] and Nano-Evaluris [82] use SDS data combined with data from occupational exposure monitoring, protective measures, production rates, and frequency of use to obtain task-specific risk assessments for nanoparticles. The French agency for food, environmental, and occupational health and safety (ANSES) considers the physical form (solid, liquid, powder, or aerosol) of a manufactured nanomaterial to determine its exposure/emission potential [83]. Among CBs, Nanotool v2.0 was the tool chosen for this study. Tables 7–9 show the results of applying this tool for the manual loading of the metal powder (cobalt and chromium) on the EOS 290 M (O1), the removal and cleaning of the final product, and the recovery of unused metal powder (nickel and chromium) inside the renAM 500 Q operating area by manually dragging the powder into the perforated grill around the building plate (O3), respectively.

**Table 7.** Severity scores of O1 and O3.

| CB Factors | O1 | O3 |
|---|---|---|
| PM OEL | 500 µg/m$^3$ * | 200 µg/m$^3$ *** |
| PM Carcinogenicity | Yes (Carc. 1B, H350) ** | Yes (Carc. 2, H351) ** |
| PM Reproductive toxicity | Yes (Repr. 1B, H360F) ** | No |
| PM Mutagenicity | Yes (Muta. 2, H341) ** | No |
| PM Dermal toxicity | Yes (Skin Sens., H317) ** | Yes (Skin Sens., H317) ** |
| PM Asthmagen | No | No |
| NM Surface chemistry | Unknown | Unknown |
| NM Particle shape | Unknown | Unknown |
| NM Particle diameter | Unknown | Unknown |
| NM Solubility | Unknown | Unknown |
| NM Carcinogenicity | Unknown | Unknown |
| NM Reproductive toxicity | Unknown | Unknown |
| NM Mutagenicity | Unknown | Unknown |
| NM Dermal toxicity | Unknown | Unknown |
| NM Asthmagen | Unknown | Unknown |
| Severity Score/Band | 71–High | 58.5–High |

* OEL D. Lgs. 81/2008, Allegato XXXVIII ** Data from safety data sheet *** TLV-TWA ACGIH [44].

The two tasks that were analysed with Nanotool showed a risk level equal to RL3, highlighting the hazardousness of metal AM and the need to contain these operations and for occupational safety specialist evaluations (Tables 7–9). The current controls were in line with what was recommended by the CB Nanotool.

**Table 8.** Probability scores of O1 and O3.

| CB Factors | O1 | O3 |
|---|---|---|
| Estimated amount of material used | >100 mg | >100 mg |
| Dustiness/mistiness | High | High |
| Number of employees with similar exposure | 1–5 | 1–5 |
| Frequency of operation | Daily | Weekly |
| Duration of operation | 30–60 min | 30–60 min |
| Probability Score/Band | 75–Likely | 70–Likely |

**Table 9.** Overall risk level without Controls of O1 and O3.

| | O1 | O3 |
|---|---|---|
| Severity Score/Band | 71–High | 58.5–High |
| Probability Score/Band | 75–Likely | 70–Likely |
| Overall Risk Level without Controls | RL3 | RL3 |

CB Nanotool v2.0 was applied because at the initial stages of the collaboration with Baker Hughes, it was necessary to assess the state of the design conditions; moreover, not all of the information was available, and it was not possible to carry out all of the simulations that would be necessary to be able to apply a dynamic evaluation. In addition, it should be considered that dynamic assessment based on machine learning requires a subjective analysis that must be carried out in collaboration between experts and Baker Hughes. Choosing incorrect or inadequate indicators could lead to overfitting; it follows that the choice and customization of a predictive model with a precise purpose should be carried out carefully and accurately by adopting the appropriate metrics, tolerance, and criteria [57].

The Baker Hughes personnel at the Florence plant were always well aware of the danger linked to metal AM: biological monitoring had been carried out on the workers exposed to metal powder in the additive laboratories since 2010 (data not shown). The urinary analysis results of the AM operators established that there were not significant increases in the amounts of chromium, cobalt, and nickel compared to the Italian reference values for metals in urine (LISTA SIVR 2017), as observed in other similar studies [46]. Baker Hughes pays close attention to biological monitoring: the objective of this biological monitoring is the results of the exposed and not exposed workers, such as those working the offices, should be comparable. Baker Hughes' policy regarding worker health is that no AM operator should show any health effect linked to working in the laboratory. To better understand and evaluate exposure in the workplace, during the years in which biological monitoring was implemented, increased attention was paid to the surveys conducted with the operators concerning their diet, drinking water, and the leisure activities outside of work that could interfere with the metal values in their urine.

The proposed environmental and biological monitoring strategy led by the Baker Hughes Florence plant would be an operative protocol to assess the risks associated metal AM and to manage metal AM in standardised conditions. In fact, as often happens with new technologies, nowadays, standards development organisations are involved in processing the knowledge gained from experience in working with AM technologies to develop new operative standards. The America Makes and ANSI Additive Manufacturing Standardization Collaborative has developed a standardization roadmap for additive manufacturing [84] that describes all of the existing AM-related standards as well as all of the AM standards that are under development, assesses the gaps in current standards, and prioritises specific areas where additional standard development efforts are needed [85]. In addition, the Food and Drug Administration (FDA) and the National Aeronautics and Space Administration (NASA) have proposed guidelines related to specific field of applications for AM via Technical Considerations for Additive Manufactured Medical Devices [86]

and Standard MSFC-STD-3716 [87], respectively. The Verein Deutscher Ingenieure (VDI) recently released standard 3405, which was introduced to deal with metal AM processes using LB-PBF technologies [88]. Furthermore, the ad hoc "Safety Issue" group of the ISO proposes safety-relevant standards to Technical Committee (TC) 261 "Additive Manufacturing" of the ISO [89]. In the operative protocols for AM, attention must also be directed towards special protective measures and equipment, especially if the activities require the use of hazardous substances that are carcinogenic, mutagenic, or toxic for reproduction (category 1A and 1B). A correct decision on the necessary protective measures fulfilling the requirements of the European Standard EN 482:2021 can be made after standardised workplace measurements using analytical methods with sufficiently long sampling durations (>180 min) have been carried out.

## 4. Conclusions

The momentum of the advances of AM metal technology is rapidly changing and moving toward the production of high-value components. AM technologies have demonstrated notable potential applications across many industries, including in the aerospace, biomedical, and oil and gas industries. The investigations performed in this measurement campaign show that the PB-AM processes in the Baker Hughes plant in Florence are in accordance with main rules and standards and that the already realised protective solutions are adequate. No operators have exceeded the OEL for inhalable and respirable dust, and concerning the UFPs, the AM working procedures do not result in a notable increase compared to other occupational scenarios. Therefore, no supplementary protective measures are needed at the moment if the process conditions do not change significantly. Regarding the qualitative risk assessment, the AM operations show a risk level that requires containment and the evaluation of an occupational safety specialist.

In the future, more monitoring campaigns must be carried out to create a comprehensive database of operative AM protocols. Moreover, the health and safety mindset should evolve to obtain more information and all of the quantitative data necessary for dynamic risk assessment and emergency management. This more holistic approach would result in the production of a learning model that could be an essential tool in occupational health and safety.

The obtained results will help us to elaborate upon the standardised working protocols and control guidance sheets for AM processes. The AM industry is paying more attention to proposing safer operational solutions; we can expect more attention to paid to the problems related to powder removal and the post-processing of AM. At present, many of the tasks that take place during these steps are still manual, but the growing automation of these tasks could have a positive effect on exposure levels in AM work areas.

**Supplementary Materials:** The following supporting information can be downloaded at https://www.mdpi.com/article/10.3390/safety8020032/s1, Figure S1: Particulate matter fractions for O1; Figure S2: Particulate matter fractions for O2; Figure S3: Particulate matter fractions for O3; Figure S4: Particulate matter fractions for O4; Figure S5: Particulate matter fractions for O5; Figure S6: Particulate matter fractions for O6; Figure S7: Inhalable, thoracic, and respirable fractions for O1; Figure S8: Inhalable, thoracic, and respirable fractions for O2; Figure S9: Inhalable, thoracic, and respirable fractions for O3; Figure S10: Inhalable, thoracic, and respirable fractions for O4; Figure S11: Inhalable, thoracic, and respirable fractions for O5; Figure S12: Inhalable, thoracic, and respirable fractions for O6; Figure S13: LDSA for O1; Figure S14: LDSA for O2; Figure S15: LDSA for O3; Figure S16: LDSA for O4; Figure S17: LDSA for O5; Figure S18: LDSA for O6.

**Author Contributions:** Conceptualization, S.D., G.C. and L.T.; methodology, G.C. and L.T.; software, G.C. and L.T.; validation, S.D.; formal analysis, G.C. and L.T.; investigation, S.K., F.P. and M.R.; resources, E.B. and D.S.; data curation, E.B. and D.S.; writing—original draft preparation, S.D., G.C. and L.T.; writing—review and editing, G.C., L.T. and S.K.; visualization, N.M.; supervision, S.K., F.P. and M.R.; project administration, G.A. and N.M. All authors have read and agreed to the published version of the manuscript.

**Funding:** This research received no external funding.

**Institutional Review Board Statement:** Not applicable.

**Informed Consent Statement:** Not applicable.

**Data Availability Statement:** Not applicable.

**Conflicts of Interest:** Among the authors, Simon Kemble, Fabrizio Paone, and Massimiliano Rigacci are employed by Baker Hughes Florence Plant. The monitoring object of the paper was part of a research project between the Department of Experimental and Clinical Medicine and Baker Hughes.

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
