# Peer review of "A Qualitative and Quantitative Occupational Exposure Risk Assessment to Hazardous Substances during Powder-Bed Fusion Processes in Metal-Additive Manufacturing"

_safety_

Round 1
Reviewer 1 Report
For metal additive manufacturing, it is no doubt that risky evaluation is an important topic. However, this paper is more of technical report than a scientific paper. The authors just describes the layout method of workshop environment and corresponding data acquisition process, and says nothing about risky evaluation method, while theoretical innovation could not be found.
Author Response
Dear Reviewer,
Thanks for your comments. The goal of our work was to develop a strategy for monitoring occupational exposure to metal powders in a metal additive manufacturing laboratory. In doing this, in addition to applying one of the most recent technologies for the real-time determination of micro and nanometric particulates, we have considered the need to apply one of the preliminary risk assessment methods, Control Banding, to be able to provide a simple instrument for the transposition of the proposed strategy in other similar scenarios.
Kind regards,
Reviewer 2 Report
The study is very interesting and actual. The paper may provide high interest for broader auditorium working on chemical risk management in the companies and workplaces of different industries.
Some notes:
- In line 96: “nazo- “
- In the section between lines 152-153 please add full names to the methods not only the acronyms.
- The same lines: it would be recommended to add references to the named methods. In some cases, the methods are rather specific and not directly attributed to the workplace and technology processes discussed in this paper (for example, TEARR is not recognised as a tool, the Worst-Case Definition is well known environmental hazard assessment method.) Some additional evaluation of these methods should be provided.
- Line: 185 m2 – superscript
- 1 It is recommended to use some colours, because the differences of patterns are rather similar (it would be for the reader better to change not only the pattern, but to apply some background colouring (as this is Online accessible journal, and colouring is recommended) It is also recommended to maybe add names of the machines not only to add brand names in the figure with little explanation in the figure caption.
- It seems from the data that draught risk may be also a potential problem in the case of such velocities. Did you measured the air flow that it does not affect the worker or the distribution of the contaminated air?
- In the fact it would be also good to show the workplaces and the movement routes in order to emphasize the concerns of other physical factor impact.
- Is there any information about filter system monitoring and frequencies of filter quality monitoring?
- Line 208: PPE – personal protective equipment
- Line 243: iCAPQ ICP/MS: iCAP Q ICP-MS
- Line 256: please concretise the method used for metal concentration determination in urine. Did authors had developed method for Me analysis in dust samples and urine by ICP-MS or some already reported methods were adapted. In both cases, information should be added.
- Line 258: please add the reference to the method description. Is this the original method: Paik SY, Zalk DM, Swuste P. (2008) Application of a pilot control banding tool for risk level assessment and control of nanoparticle exposures. Ann Occup Hyg; 52: 419–28?
- Table 1: please check, whether there is everything ok with risk levels? It would be recommended to recheck the levels as it is not clear whether the different factors could be analysed as the levels are commonly attributed to increasing hazard risks, in the case of named factors, their combinations also should be considered.
- BCK – acronym should be specified
- In Figures 2-7, the tittles should be removed. In the figure captions, the risks O! -O6 should be specified (it is hard to analyse data and concentrate on the risk description.
- The discussion should be improved. It is hard to concentrate on data when all the Figures are repeatedly shown without included elevation of the data and explanation of these results.
Author Response
Dear Reviewer,
please see the attachment. Thank you for your comments.
Kind regards,

Reviewer 3 Report
|
|
|||
Dear Authors, first of all let me express my compliments for the interesting subject of your research. I do not have serious negative considerations about your paper in its structure, results showed and analysis about them.
I think the paper is interesting and it could be considered for publication. However, this paper can be further improved in the following points.
- the English level should be improved.
- the "Introduction" should be shorten or may be divided in two section (introduction and literature review);
- "future works" should be better explained
Please improve english and references, e.g.:
- Reliability estimation of reinforced slopes to prioritize maintenance actions
Bahootoroody, F., Khalaj, S., Leoni, L., ...Di Bona, G., Forcina, A. International Journal of Environmental Research and Public Health, 2021, 18(2), pp. 1-12, 373;
- Systematic Human Reliability Analysis (SHRA): A New Approach to Evaluate Human Error Probability (HEP) in a Nuclear Plant
Di Bona, G. Falcone, D., Forcina, A., Silvestri, L.
International Journal of Mathematical, Engineering and Management Sciences, 2021, 6(1), pp. 345-362
- Quality Checks Logit Human Reliability (LHR): A New Model to
Evaluate Human Error Probability (HEP)
Gianpaolo Di Bona , Domenico Falcone, Antonio Forcina, Filippo De Carlo ,
and Luca Silvestri Mathematical Problems in Engineering
Volume 2021, Article ID 6653811, 12 pages https://doi.org/10.1155/2021/6653811
Reviewer 4 Report
Dear Authors,
The paper presents a topic of interest for researchers and practitioners. However, a number of improvements are needed.
- The abstract should present the need for research, the methodology, the main results obtained and the future directions of research.
- To emphasize the need for this study.
- The stages of the methodology must be presented in detail.
- To highlight the gaps filled by the present study.
- The conclusions section should be completed with a review of the study.
Round 2
Reviewer 1 Report
1. Grammar check is needed. e.g"To analyse metals in dust samples a new method has been developed, based on previous experiences and thanks to the collaboration with instrumentation specialists. " 2. Pls add the definition of risky, and explain the relationship between probability, severity and risky. 3. I still doubt about the validity of evaluation outputs, and suggest to use a learning model.Author Response
Thank you for your comments. To better explain how we have followed your recommendations, we attached your requests in this file followed by the respective answers.
- Grammar check is needed. e.g. "To analyse metals in dust samples a new method has been developed, based on previous experiences and thanks to the collaboration with instrumentation specialists. "
In the tracking version of the manuscript, you could find the correction added in the main-text.
- Pls add the definition of risky, and explain the relationship between probability, severity and risky.
We have added in the introduction the definition of risk and the relationship between probability and severity, as stated in D.Lgs 81/08, the mandatory regulation in Italy.
- I still doubt about the validity of evaluation outputs and suggest to use a learning model.
We have added the explanation of the choice for CB, instead of machine learning models, to assess the occupational risk in the study setting. We have underlined the need for a dynamic approach to risk evaluation; however, the lack of information (about the correct indicators and parameters to obtain an operational learning model) led us to use a screening, easy-to-use strategy for risk assessment.
Reviewer 4 Report
Dear Author,
I accept the present form of the paper.
Best regards,
Reviewer.